# Bacterial Community Diversity and Bacterial Interaction Network in Eight Mosquito Species

**DOI:** 10.3390/genes13112052

**Published:** 2022-11-07

**Authors:** Herculano da Silva, Tatiane M. P. Oliveira, Maria Anice M. Sallum

**Affiliations:** Department of Epidemiology, School of Public Health, University of São Paulo, São Paulo 01246-904, SP, Brazil

**Keywords:** Culicidae, bacterial network, *Wolbachia*, *Asaia*, Vale do Ribeira, 16S rRNA

## Abstract

Mosquitoes (Diptera: Culicidae) are found widely throughout the world. Several species can transmit pathogens to humans and other vertebrates. Mosquitoes harbor great amounts of bacteria, fungi, and viruses. The bacterial composition of the microbiota of these invertebrates is associated with several factors, such as larval habitat, environment, and species. Yet little is known about bacterial interaction networks in mosquitoes. This study investigates the bacterial communities of eight species of Culicidae collected in Vale do Ribeira (Southeastern São Paulo State) and verifies the bacterial interaction network in these species. Sequences of the 16S rRNA region from 111 mosquito samples were analyzed. Bacterial interaction networks were generated from Spearman correlation values. *Proteobacteria* was the predominant phylum in all species. *Wolbachia* was the predominant genus in *Haemagogus leucocelaenus*. *Aedes scapularis*, *Aedes serratus*, *Psorophora ferox*, and *Haemagogus capricornii* were the species that showed a greater number of bacterial interactions. Bacterial positive interactions were found in all mosquito species, whereas negative correlations were observed in *Hg. leucocelaenus*, *Ae. scapularis*, *Ae. serratus*, *Ps. ferox*, and *Hg. capricornii*. All bacterial interactions with *Asaia* and *Wolbachia* were negative in *Aedes* mosquitoes.

## 1. Introduction

The Culicidae family includes 3608 mosquito species (Diptera), classified into 113 genera [1]. Many species are important because they feed on human and other animals’ blood, causing a disturbance, and 5% of the known species may participate in the transmission of parasites and pathogens to humans [2,3]. Mosquitoes harbour a great diversity of fungi and bacteria. The bacterial composition of the microbiota of these invertebrates depends on several factors, such as sex, larval habitat, developmental stage, environment, and species. It can also be related to microsporidian infection [4,5,6,7]. Bacteria from the microbiota can influence mosquitoes’ physiology, metabolism, and adaptation [4]. They also play a key role in protecting against pathogens [8] and developing mosquito larvae [9,10].

Recently, based on the results of an experimental investigation, Saab et al. [5] found that *Anopheles gambiae* and *Aedes albopictus* larvae reared in the same insectary room and fed on the same food source showed intraspecies and interspecies variations in the composition of the midgut microbiota, concluding that bacterial composition can be modulated by environmental variations, species factors, and complex microbial interactions.

Despite several studies focused on microbial interactions in mosquitoes, it is still unclear how these interactions may contribute to the formation of bacterial communities in mosquitoes. There are few records of natural infection of *Wolbachia* in *Aedes aegypti* [11] and anophelines [12,13]. The absence of this bacterium in most populations of these mosquito species may be due to the presence of *Asaia* in the native microbiome, which inhibits the vertical transmission of *Wolbachia* [14,15]. *Serratia* is present in *Anopheles* mosquitoes [16,17], influencing these insects’ vector capacity [18]. In field-collected *Aedes* mosquitoes, the prevalence of this bacterium is variable. For example, *Serratia* was more abundant in *Ae. albopictus* than in *Ae. aegypti* collected in Houston, TX, United States [19]; *Serratia marcescens* was found in *Ae. albopictus* collected in Sing Bur Province of Thailand, and absent in mosquitoes of this species collected in Chumphon and Yala Provinces [20]. The occurrence of *Serratia* in *Ae. aegypti* can be low because of the competitive exclusion with *Cedecea* [21]. Interactions between bacteria, viruses, microsporidian, fungi and *Plasmodium* also changes the bacterial community [6,22,23,24,25]. Specimens of *Culex pipiens* infected with the West Nile virus showed an increased proportion of *Serratia* [26]. In addition, the endosymbiotic bacteria *Spiroplasma* sp. PL03 and *Weissela* cf. *viridescens* depends on microsporidia infection in the mosquito gut [6], and the bacterial microbiota was distinct between specimens of *An. gambiae* and *Anopheles funestus* that were positive and negative groups for *Plasmodium falciparum* [22]. In addition, in the same study, authors observed that some bacterial species such as *Asaia borgorensis*, *Burkholderia fungorum*, *Burkholderia cepacia* and *Enterobacter cloacae* were present only in females that were negative for *P. falciparum*. Balaji et al. [27] demonstrated that *Wolbachia* can influence the colonization of certain bacterial taxa by competitive interactions, such as the abundance of *Serratia* sp. in *Wolbachia*-carrying mosquitoes. The 16S rRNA data have been used to verify microbial co-exclusion/co-occurrence in different organisms [19,28,29]. Relative abundance and/or presence/absence parameters are used to verify bacterial interactions and to infer bacterial networks.

The number of studies focusing on bacteria and mosquitoes is increasing. Among these studies are: understanding the acquisition and composition of the microbiota [4,20], its relationship with vector capacity [30], its importance in the modulation of pathogen development in mosquitoes [31], and how bacterial interactions can modulate the bacterial community in the mosquito [19]. Despite this increasing interest in mosquito microbial investigations and the epidemiological importance of the Vale do Ribeira region as a potential source of sylvatic arboviruses and vertebrate reservoirs, little is known about the microbiota in mosquito populations inhabiting the southeastern Atlantic tropical rain forest, São Paulo, Brazil.

This study aims to (1) investigate the microbiota present in eight species of Culicidae collected in Vale do Ribeira, and (2) verify the bacterial communities’ composition and interactions (such as co-exclusion and co-occurrence) between these communities.

## 2. Materials and Methods

### 2.1. Mosquito Collection and Identification

Adult mosquitoes were collected in a forest preserved area neighboring the town (24°47′28.8″ S, 47°54′42.6″ W) in Vale do Ribeira, Pariquera-Açu municipality, São Paulo state, Brazil. The collections were performed using an entomological net and conducted daily from 8:00 to 14:30 h from 14 December to 17 December 2021. Mosquitoes were killed with ethyl acetate (C_4_H_8_O_2_) and immediately preserved in silica gel. Specimens were transported at room temperature to the Laboratório de Entomologia em Saúde Pública–Sistemática Molecular and kept in these conditions until identification and processing. Specimens were morphologically identified using the identification key of Forattini [2].

### 2.2. Sequencing of 16S rRNA

The mosquito’s surface was rinsed in 70% ethanol and ultrapure water. The genomic DNA of each specimen was extracted separately using the Quick-DNA Fungal/Bacterial Miniprep kit (ZymoReasearch), following the manufacturer’s instructions. The V4 region of the 16S rRNA gene was amplified using isolated DNA from each specimen. Each reaction was carried out in a final volume of 20 µL containing 1 X GoTaq^®^ Colorless Master Mix (Progema, USL), 0.3 µm of each primer (16S-V4 Forward: 5′ GTGCCAGCMGCCGCGGTAA 3′; 16S- V4 Reverse: 5′ GGACTACHVGGGTWTCTAAT 3′) [32], 8 µL of genomic DNA and ultrapure water. The PCR thermal conditions were 94 °C for 3 min, followed by 30 cycles of 94 °C for 45 s, 55 °C for 1 min, 72 °C for 1 min, and a final extension of 72 °C for 10 min. The PCR products were visualised on a 2% agarose gel stained with UniSafe Dye 0.03% (*v*/*v*) and purified with Agencourt AMPure XP magnetic beads (Beckman Coulter, Brea, CA, USA) according to the manufacturer’s recommendations. After indexing with Nextera XT Index kit (Illumina), the products were purified with magnetic beads and quantified by real-time PCR (qPCR) with the KAPA-KK4824 kit (Library Quantification kit–Illumina/Universal) following the manufacturer’s instructions. All samples were normalised to 4 nM, and an equimolar pool of DNA was prepared. Next-generation sequencing was performed on the Illumina MiSeq sequencer (Illumina, San Diego, CA, USA) using the MiSeq Reagent Micro v2 kit (300 cycles: 2 × 150 bp).

### 2.3. Processing of 16S Sequences and Taxonomic Attribution

FLASH v. 1.2.11 [33] was used to assemble Illumina paired-end reads with a minimum overlap of six base pairs. Quality control (denoising), taxonomic attribution, and diversity and abundance analyses were performed in QIIME2 v.2021-11 software [34].

The *qiime tools import* command and Casava 1.8 single-end demultiplexed format were used to import the joined sequences to QIIME2. Quality control and denoising were performed with the commands: *qiime quality-filter q-score* and *qiime deblur denoise-16S*, respectively. To taxonomic attribution was used SILVA 138 [35,36] and *qiime feature-classifier classify-sklearn*.

### 2.4. Diversity Analyses

The rarefaction curve was generated to obtain the expected number of ASVs (Amplicon Sequence Variants) in each sample for a given number of sequences, and to allow for the comparison between the richness of the samples. We used a depth value of 10,000 and *qiime diversity α-rarefaction* command to construct this curve.

Diversity metrics (α and β) were generated with *qiime diversity core-metrics-phylogenetic*, and a phylogenetic tree was constructed with *qiime phylogeny align-to-tree-mafft-fasttree.*

Shannon-Weaver indices (α diversity) were subject to Kruskal-Wallis followed by Dunn’s test, adjusted with Bonferroni method in RStudio v.1.4.1106 to verify whether samples from one species have greater α diversity than from other species.

Analysis of variance (PERMANOVA) statistical test was performed with β diversity data generated by Unifrac weighted and unweighted distances in QIIME 2. This statistical test makes it possible to verify whether bacterial diversity differs significantly between mosquitoes of different species.

### 2.5. Microbiome Composition Analysis

Microbiome composition analysis (ANCOM) was performed in QIIME 2 using ASV table, SILVA taxonomy and the commands: *qiime taxa collapse*, *qiime composition add-pseudocount*, and *qiime composition ancom*. This study makes it possible to verify whether any ASV is more abundant in a particular mosquito species than in another.

### 2.6. PCoA and Heatmap

Data of the weighted and unweighted Unifrac phylogenetic distance matrices generated in QIIME2 were used to perform Principal Coordinate Analysis (PCoA) in RStudio v.1.4.1106. PCoA images were generated with tidyverse and qiime2R packages available in RStudio. These images allow for visualizing the distance between the bacterial communities of each sample. Heatmaps were obtained in RStudio v.1.4.1106 employing data of ASV table and taxonomy. The images generated allow for verifying the abundance of each taxon per sample.

### 2.7. Bacterial Interaction Network

The ASV table was changed as described in Hegde et al. [19] and was then used for the bacterial interaction (co-exclusion/co-occurrence) analyses. We performed the modifications to filter the ASV table data: (1) ASVs with readings lower than 0.1% of the total number of readings from all samples were removed; (2) the remaining ASVs were combined according to the common taxonomy assignments’ lowest until genus level; (3) for each species, a relative abundance table was generated by dividing the number of sequence reads of each bacterial taxon per the total, initialing the number of sequence reads of each sample and then multiplying by 100.

After data normalization, a Spearman correlation matrix was generated. This matrix and analysis were performed for each species separately and carried out with igraph, Hmisc and Matrix packages available in RStudio. Spearman correlations with values of r ≤ 0.75 were discarded, as well as with *p* ≥ 0.05. The non-discarded data were used to infer a bacterial interaction network, with the blue lines corresponding to a positive correlation, whereas red lines corresponded to a negative.

## 3. Results

### 3.1. 16S Sequences Data

One-hundred-and-eleven female mosquitoes were used to obtain bacterial 16S rRNA sequences. These samples correspond to the following species: *Ae*. *scapularis* (08), *Ae*. *serratus* (06), *Hg*. *capricornii* (07), *Hg*. *leucocelaenus* (14), *Ke*. *cruzii* (15), *Ps*. *ferox* (12), *Sa*. *conditus* (30) and *Wy*. *confusa* (19) (Appendix A).

A total of 9,647,379 (R1 or R2) raw reads were generated in the NGS. These reads varied between 58,143 and 123,070 in the samples (Appendix A). After joining forward and reverse reads and filtering steps, 2,090,703 sequences were retained for analyses (Appendix A).

### 3.2. Bacterial Diversity

A total of 2617 ASVs were identified in the samples. *Ae. scapularis* showed 530 ASVs; *Hg. leucocelaenus*, 310; *Hg. capricornii*, 334; *Ps. ferox*, 874; *Ae. serratus*, 504; *Sa. conditus*, 1014; *Wy. confusa*, 623; and *Ke. cruzii*, 546 (Appendix A). *Proteobacteria* was the predominant phylum, followed by *Firmicutes* and *Actinobacteriota* (Figure 1 and Appendix A). *Wolbachia* was the predominant genus in *Hg. leucocelaenus*. *Afipia*, *Acinetobacter*, and *Asaia* genera were abundant in most of the species analysed (Figure 2 and Appendix A).

### 3.3. Diversity Analysis

The number of sequences retained for the analyses was sufficient to infer the abundance of the bacterial community in each sample (Appendix A). All samples were normalised to calculate the diversity metrics. The cut-off value for normalisation was the lowest number of contigs found in the sequenced samples. This value corresponds to 8857 sequences, as shown in Appendix A.

Shannon-Weaver indices ranged from 0.607 to 6.355 between samples (Appendix A). These indices did not show a normal distribution (Shapiro test; W = 0.97394, *p* = 0.028), and then the Kruskal–Wallis test (χ^2^ = 24.393, *p* = 0.0009) followed by Dunn’s test adjusted with the Bonferroni method was performed to know which species differed in α diversity. The following groups showed significant differences in the Shannon indices: *Hg. leucocelaenus*–*Ke. cruzii* (*p* = 0.002); *Hg. leucocelaenus*–*Sa. conditus* (*p* = 0.002) and *Hg. leucocelaenus*–*Wy. confusa* (*p* = 0.026).

The β diversity was calculated for each species and visualised in the PCoAs (Figure 3). Significant differences in bacterial composition were verified with PERMANOVA analysis using unweighted and weighted Unifrac distances data (Appendix A). Significant differences in both PERMANOVA analyses were found between (1) *Ae. scapularis* and *Hg. leucocelaenus*, *Ke. cruzii*, *Sa. conditus*, and *Wy. confusa*; (2) *Ae. serratus* and *Hg. capricornii*, *Hg. leucocelaenus*, *Ke. cruzii*, *Sa. conditus*, and *Wy. confusa*; (3) *Hg. capricornii* and *Ke. cruzii*, *Sa. conditus* and *Wy. confusa*; (4) *Hg. leucocelaenus* and *Ke. cruzii*, *Sa. conditus*, *Wy. confusa*, and *Ps. ferox*; (5) *Ke. cruzii* and *Sa. conditus* and *Ps. ferox*; (6) *Ps. ferox* and *Sa. conditus* and *Wy. confusa*; and (7) *Sa. conditus* and *Wy. confusa* (Appendix A).

### 3.4. Microbiome Composition Analysis and Heatmap

In the ANCOM analysis, it was possible to verify that *Wolbachia* was more abundant in *Hg. leucocelaenus*, while *Afipia* and *Asaia* genera were more abundant in *Ke. cruzii* and *Ae. serratus*, respectively (Appendix A).

Bacteria of the *Proteobacteria* phylum showed the greatest number of sequences in all species (Appendix A). Figure 4 shows that a larger number of *Wolbachia* sequences was found in *Hg. leucocelaenus*, whereas *Afipia* was more abundant in *Ke. cruzii*, *Sa. conditus* and *Wy. confusa.*

### 3.5. Bacterial Interaction Network

Interactions between bacterial communities in the samples were verified based on the 16S sequences. The ASV table–filtered and normalized (Appendix A) of each species–was used for bacterial interaction analysis and then interaction data was used to infer bacterial interaction networks (Figure 5 and Figure 6). Bacteria found in *Ke. cruzii*, *Sa. conditus*, *Hg. leucocelaenus*, and *Wy. confusa* showed fewer interactions than those present in *Ae. scapularis*, *Ae. serratus*, *Hg. capricornii*, and *Ps. ferox*. Positive bacterial correlations were found in all species analysed, while negative correlations were observed in *Hg. leucocelaenus*, *Ae. scapularis*, *Ae. serratus*, *Ps. ferox*, and *Hg. capricornii* (Figure 5 and Figure 6, Appendix A). The bacteria that showed more interactions were *Shingobium*, Unclassified *Pseudonocardiaceae*, *Afipia*, *Delftia*, and *Shingomonas* in *Aedes* samples; *Shingomonas*, *Afipia*, *Methylobacterium-Methylobacterium*, *Pseudomonas*, *Puia*, *Acidibacter*, and *Afipia* in *Hg. capricornii*; and *Shingomonas*, *Methylobacterium-Methylobacterium*, *Acidibacter*, *Afipia*, and *Pseudomonas* in *Ps. ferox*. All bacterial interactions with *Asaia* and *Wolbachia* were negative in the *Aedes* species (Figure 5).

## 4. Discussion

Culicidae mosquitoes have a broad geographic distribution. Although insects of this family have great epidemiological importance in the transmission of pathogens [37,38] and studies show the influence of the microbiota in this transmission [31], little has been studied about bacteria present in mosquitoes collected in Vale do Ribeira, Brazil [39]. This region is known to harbor a great diversity of vector mosquitoes, to register autochthonous malaria cases (bromeliad-malaria) [40], and to be highly vulnerable to sylvatic yellow fever [41]. Thus, this study contributes to a better understanding of the bacterial communities in eight mosquito species found in areas of the Vale do Ribeira, southeastern Atlantic tropical rain forest.

Because the mosquito species analysed in this study have great public health importance as potential vectors of arboviruses, the knowledge of bacterial communities present in these insects is essential for further investigating the bacteria-pathogen interactions and their influence on the ability of these species to be infected and transmit parasites. *Ae*. *scapularis* has a wide distribution and can be a vector for Yellow Fever Virus (YFV), Venezuelan Equine encephalitis, and *Wuchereria bancrofti* [42,43,44]. *Ae*. *serratus* is considered a secondary vector of both the Ilheus virus [45] and the YFV [37,38]. *Hg. leucocelaenus* is the primary vector of sylvatic YFV in the New World and was involved in Brazil’s greatest yellow fever outbreak between 2016 and 2018 [38]. *Ke*. *cruzii* is the primary vector of *Plasmodium* sp. in Atlantic Forest (bromeliad malaria) [46,47], and it was found to be naturally infected with the Zika virus [48]. *Ps*. *ferox* has already been found to be naturally infected with the Rocio virus in Vale do Ribeira region [49], with YFV in the municipality of Urupês, Ribeirão Preto region, São Paulo state [41]. *Wy*. *confusa* from Capivari-Monos EPA (São Paulo municipality) was infected with the Zika virus [48].

Many factors can modulate mosquitoes’ bacterial composition, including the mosquito species [50] and complex microbial interactions. In addition, the mosquito developmental stage, geographic location of the samples, the collection period, and adult sex can affect the bacterial composition and their interactions [5]. In this study, all specimens analysed were females collected in the same period and geographical location. Thus, the data obtained are a rich source of information, as they show the scenario of bacterial diversity and the networks of interactions that occur in a given species while in contact with nature.

Statistical tests showed a significant impact of the species on the α diversity of the mosquito, with an effect verified between *Hg. leucocelaenus* and *Ke. cruzii* and *Sa. conditus* and *Wy. confusa*. Permanova analyses showed that most mosquito species studied differed in their bacterial communities. These results corroborate other studies in the literature [5,51,52], and indicate that the species can influence the formation of bacterial composition in mosquitoes. In addition, some species are associated with distinct phytotelmata habitats, such as *Ke. cruzii* linked to bromeliad phytotelmata; *Wy. confusa*, *Sabethes*, and *Haemagogus* with phytotelmata provided by bamboo and tree holes; and other species are relations with temporary ground pools, such as *Ae. scapularis*, *Ae. serratus* and *Ps. ferox.* Although all specimens have been collected in the same macroregion, the larval habitats (microregions) are ecologically distinct. As bacterial communities present in immature habitats can contribute to bacterial composition in adult mosquitoes [53], the bacterial differences observed in this study can be accounted for not only by the species differences, but also by distinct larval habitats.

There are several interactions between microorganisms, such as mutualism, competition, and commensalism [28]. Increased interactions between bacterial microbiota can alter gene expression in the microbial community, the metabolism, and ecological interactions between species [54,55]. This study exploited bacterial interaction networks using the 16S sequence data from 111 specimens of eight mosquito vector species. Among the species studied, *Ae. serratus* showed the highest interactions between bacterial communities, followed by *Hg. capricornii*, *Ae. scapularis*, and *Ps. ferox*.

*Afipia* was found in all mosquito species, being found predominantly in *Wy. confusa*, *Ke. cruzii*, and *Sa. conditus*. Despite being in lower abundance in *Ae. scapularis*, *Ae. serratus*, *Hg. capricornii*, and *Ps. ferox*, in these species, *Afipia* interacted with at least six other bacterial genera. *Afipia* is a Gram-negative rod bacterium of the phylum *Proteobacteria*. The genus was described in 1991 [56], and one species (*Afipia felix*) appears to be related to cat scratch disease [57]. To date, there was no record of *Afipia* in mosquitoes.

*Acinetobacter* was found in all mosquito species studied. Species of this genus may be involved in blood digestion and parasite–vector interactions in *Ae. albopictus* [58]. Bacteria of the genus *Delftia* were found in females of *Nyssorhynchus darlingi* not infected with *Plasmodium* [59] in other Culicidae species [60]. The bacterial interaction analyses showed a negative correlation between *Delftia* and *Asaia* in *Aedes* mosquitoes. The negative interaction between these bacterial groups needs further investigation to verify if the co-exclusion hypothesis of these bacterial genera can occur in other *Aedes* mosquitoes.

*Asaia* is frequently found in *Aedes* and *Anopheles* mosquitoes [58,61], and it can inhibit the development of *Plasmodium* in female mosquitoes [62]. In addition, *Asaia* can diminish the longevity of infected males of *Anopheles stephensi* [63]. It is noteworthy that *Asaia* and *Wolbachia* showed only negative interactions with other bacterial genera in the *Aedes* samples analyzed in this study. Our findings corroborate the results of an investigation to verify the occurrence of reciprocal negative interference between these bacterial groups to colonise the gonads [15]. Although the results of bacterial interaction analyses did not show the mutual exclusion observed by Rossi and colleagues, the current study found differences between the number of sequences of these two genera in the *Aedes* species from Vale do Ribeira. Few *Wolbachia* sequence readings were found in *Ae. scapularis* and *Ae. serratus*, while *Asaia* readings were more abundant in both species. Although the *Wolbachia-Asaia* mutual exclusion hypothesis has been found in *Hg. leucocelaenus* specimens from Vale do Ribeira [39], the same effect was not verified in the females analyzed for this study, collected in the same geographical region.

*Wolbachia* infection in *Ae. aegypti* causes upregulation in the transcription of genes related to reduction–oxidation reactions and immunity. This leads to an increase of reactive oxygen species and induction of oxidative stress in the host and showing that to favor infection, the bacterium can manipulate the host’s defense system [64].

Many of the *Wolbachia* strains induce cytoplasmic incompatibility (CI), resulting in lethality of embryos generated from females not infected with *Wolbachia* or infected with a different strain present in the male [65]. It manipulates the reproduction of the host and facilitates the spread of this bacterium, since infected females have this reproductive advantage over uninfected ones. Many studies use CI as a possible method of biological control [66]. In addition, this bacterium can reduce the infection of the dengue virus in the salivary glands of *Ae. albopictus* [67] and to inhibit yellow fever virus replication in *Ae*. *aegypti* [68]. Thus, considering that this bacterium was abundant in samples of *Hg. leucocelaenus* and this species of mosquito is considered the primary vector of sylvatic yellow fever virus in the south-eastern region of Brazil [38], further studies need to be carried out to know the *Wolbachia* strains present in these mosquitoes, and to know if the existing strains can harm the spread of the yellow fever virus.

*Serratia* bacteria can modulate the vector competence of populations of *Ae. aegypti* for the Zika virus by interfering with mosquito salivation [69] and interfering in the vectorial capacity of other mosquito species [70]. In this study, *Serratia* 16S readings were found in one female of each species, *Sa. conditus* and *Ps. ferox*. The competitive exclusion between *Serratia* and *Cedecea* was verified in *Ae. aegypti* co-infected with these bacteria [21]. No reading of the *Cedecea* was recovered from the specimens of the species of the Aedini tribe from Vale do Ribeira. Consequently, the absence of *Serratia* in *Aedes* and *Anopheles* mosquitoes cannot be explained by the exclusion effect between *Serratia* and *Cedecea* symbionts. It is likely that other bacteria can be responsible for the competitive exclusion with *Serratia*.

Our findings uncovered that the analyzed species exhibited clear differences in their microbiota composition based on the α and β diversity indices. Furthermore, some bacterial interactions in this study corroborated previous findings in the literature, while other interactions will need further investigation. In addition, it will be important to verify if the interactions are observed in different spatial and temporal scales in subpopulations of a mosquito species.

## 5. Conclusions

Here we uncover variations in microbial composition among different Culicidae species. *Wolbachia* and *Asaia* are predominant in species of the Aedini tribe, such as *Hg. leucocelaenus* and *Aedes* mosquitoes. The bacterial interaction network was verified for each mosquito species showing that *Shingobium*, Unclassified *Pseudonocardiaceae*, *Afipia*, *Delftia*, and *Shingomonas* had more bacterial interactions in the *Aedes* specimens. Bacterial interactions with *Asaia* and with *Wolbachia* were negative in *Aedes* mosquitoes.

## Figures and Tables

**Figure 1 genes-13-02052-f001:**
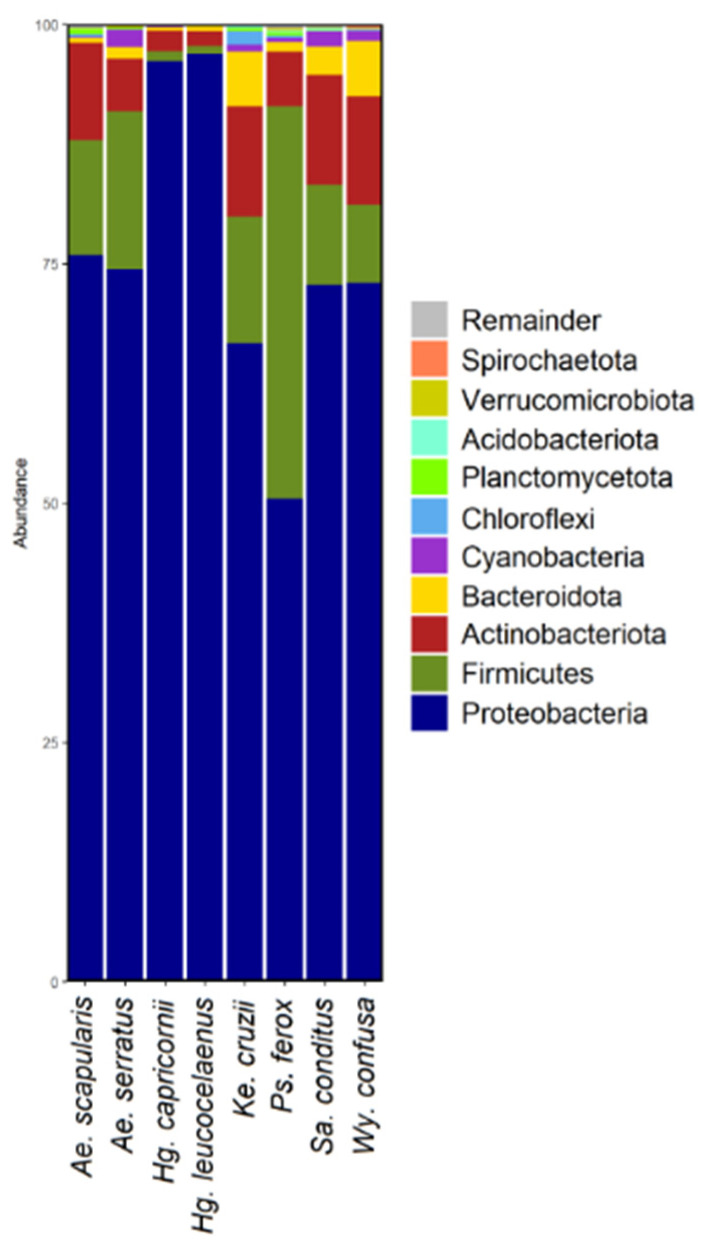
Bar graph depicting the bacterial amplicon sequence variants (ASV) composition at the highest taxonomic level (phylum) for each mosquito species.

**Figure 2 genes-13-02052-f002:**
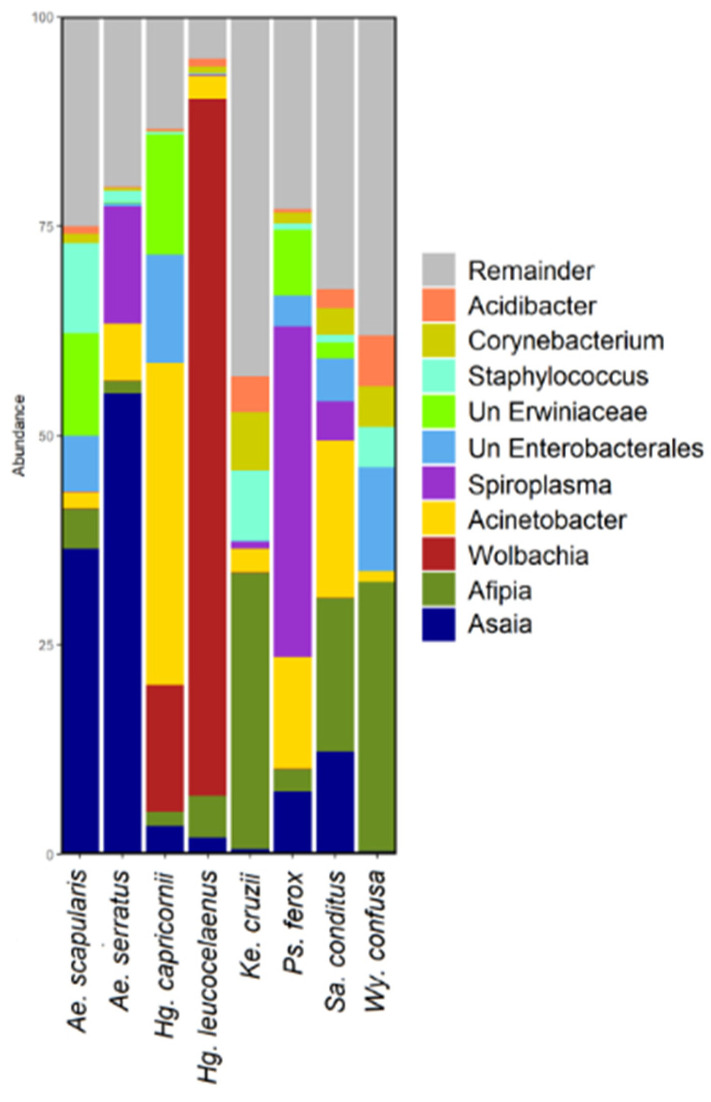
Bar graph depicting the bacterial amplicon sequence variants (ASV) composition at the bottom taxonomic level (genus) for each mosquito species. Remainder (gray bars) corresponds to genera that had 2% or less of the total contigs analyzed.

**Figure 3 genes-13-02052-f003:**
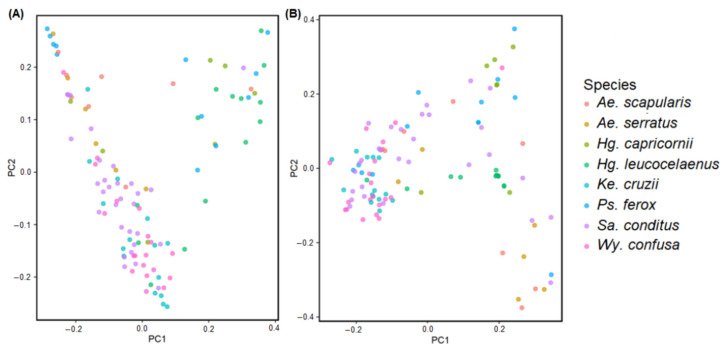
Principal Coordinate Analysis (PCoA) plots β diversity differences between the mosquito species. (**A**) PCoA using unweighted distance data. (**B**) PCoA using weighted distance data.

**Figure 4 genes-13-02052-f004:**
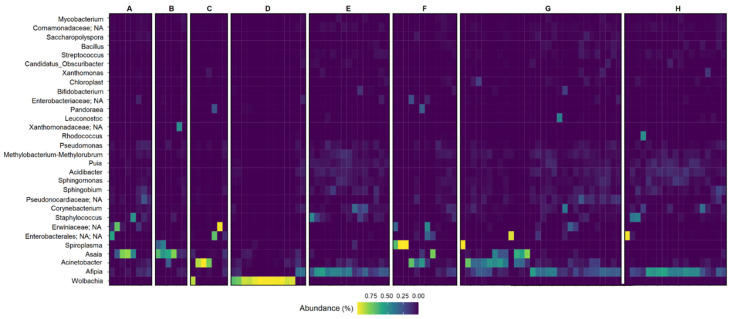
Heatmap of bacterial sequences with taxonomic assignment to genus level in each mosquito sample. Each row represents a bacterial taxon and each column corresponds a mosquito sample. Relative abundance data are assigned colors across a gradient from yellow (higher bacterial abundance) to purple (lowest bacterial abundance). (**A**) corresponds to samples of the species *Ae. scapularis*; (**B**) corresponds to samples of the species *Ae. serratus*; (**C**) corresponds to samples of the species *Hg. capricornii*; (**D**) corresponds to samples of the species *Hg. leucocelaenus*; (**E**) corresponds to samples of the species *Ke. cruzii*; (**F**) corresponds to samples of the species *Ps. ferox*; (**G**) corresponds to samples of the species *Sa. conditus* and (**H**) corresponds to samples of the species *Wy. confusa*.

**Figure 5 genes-13-02052-f005:**
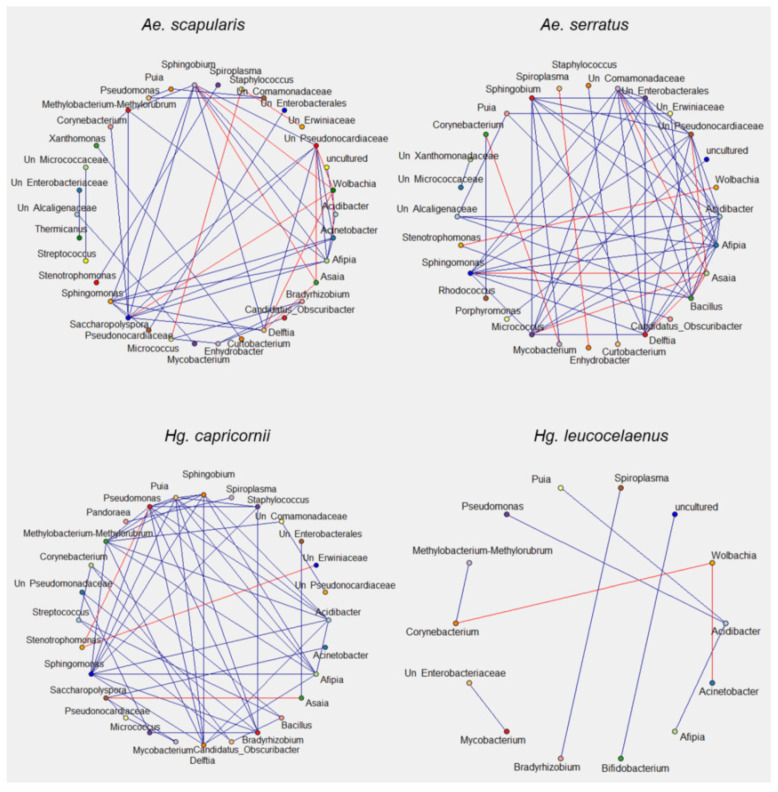
Bacterial interaction network in *Ae. scapularis*, *Ae. serratus*, *Hg. capricornii* and *Hg. leucocelaenus*. Each colored circle represents a bacterial taxon. Blue line corresponds to positive correlation between taxa. Red line represents negative correlation between taxa. Correlations were obtained by Spearman and only considered correlations with r > 0.75 and *p* > 0.05.

**Figure 6 genes-13-02052-f006:**
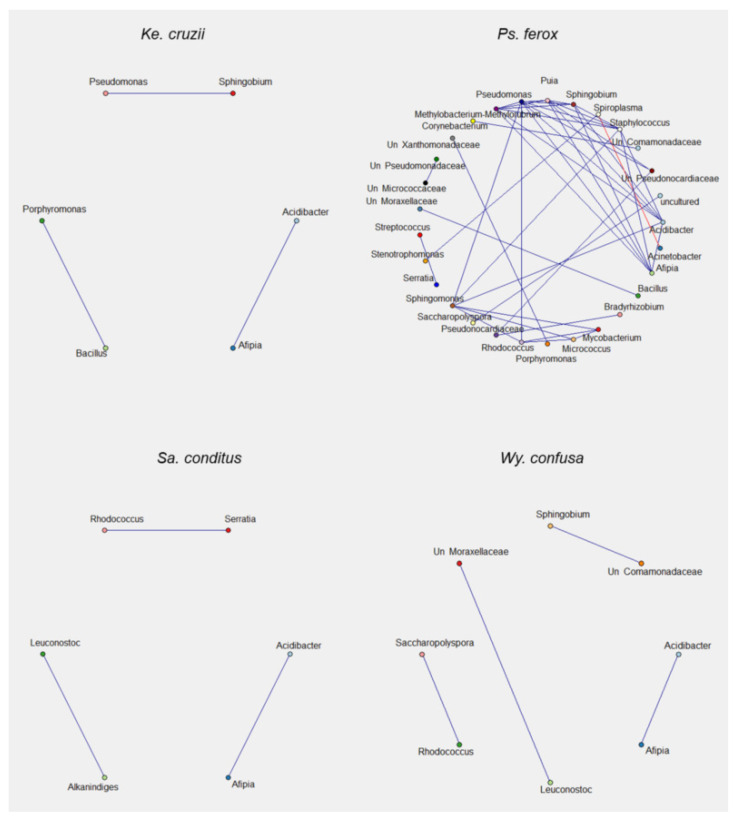
Bacterial interaction network in *Ke. cruzii*, *Ps. ferox, Sa. conditus and Wy. confusa*. Each colored circle represents a bacterial taxon. Blue line corresponds to positive correlation between taxa. Red line represents negative correlation between taxa. Correlations were obtained by Spearman and only considered correlations with r > 0.75 and *p* > 0.05.

## Data Availability

Sequence data (individual fastq files) are available from the NCBI Sequence Read Archive under accession PRJNA857687 and also in the Zenodo repository (https://doi.org/10.5281/zenodo.7300402).

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
