# Peer review of "Bacterial Community Diversity and Bacterial Interaction Network in Eight Mosquito Species"

_genes, 2022, doi:10.3390/genes13112052_

Round 1

Reviewer 1 Report

The manuscript presents a research that has a moderate but substantial level of originality that resides essentially in the mosquito species used for the study of their microbiota.
Basically it is a good descriptive work regarding bacterial community diversity (it could be useful to refer to and cite the work of Mancini et al of 2018 on BMC Microbiology).
What does not convince me is the part concerning the "Interaction Ntework". Speaking of interactions I would have expected a study on molecular / cellular interactions between different bacteria; instead here we are only talking about the co-presence or absence of different bacterial species with reference to a specific taxon of mosquitoes.
Considering that the NGS does not produce quantitative data and that mosquitoes have not been analyzed with reference to specific tissues / organs, I believe it is deeply wrong to speak of interactions and networks.

The discussion is really too verbose and in the first part it seems more like an introduction.
There are some spelling errors (i.e. lane 146 species instead of species)

Author Response

Reviewer 1

The manuscript presents a research that has a moderate but substantial level of originality that resides essentially in the mosquito species used for the study of their microbiota.

Basically it is a good descriptive work regarding bacterial community diversity (it could be useful to refer to and cite the work of Mancini et al of 2018 on BMC Microbiology).

Reply: Authors appreciated your time and effort of in reviewing this manuscript. Thank you for this observation. Mancini et al. 2018 is now in line 432.

What does not convince me is the part concerning the "Interaction Ntework". Speaking of interactions I would have expected a study on molecular / cellular interactions between different bacteria; instead here we are only talking about the co-presence or absence of different bacterial species with reference to a specific taxon of mosquitoes.

Reply: Thank you for your comment. Studies published in the literature showed some bacteria can influence the formation of bacterial communities, either by favoring or harming the colonization of certain taxa, such as Wolbachia and Asaia, Wolbachia and Serratia. In the literature, we found co-exclusion/co-occurrence analyzes as the method to verify the interaction between bacterial taxa (Chaffron et al., 2010; Barberán et al., 2012; Faust et al., 2012). Based on these interactions, it inferred microbial interaction networks. Thus, it is possible to observe pairs of taxa that have co-exclusion or co-occurrence relationships.

Chaffron, S., Rehrauer, H., Pernthaler, J., & von Mering, C. (2010). A global network of coexisting microbes from environmental and whole-genome sequence data. Genome research, 20(7), 947–959. https://doi.org/10.1101/gr.104521.109

Barberán, A., Bates, S., Casamayor, E. et al. (2012) Using network analysis to explore co-occurrence patterns in soil microbial communities. ISME J 6, 343–351. https://doi.org/10.1038/ismej.2011.119.

Faust, K., Raes, J. (2012) Microbial interactions: from networks to models. Nat Rev Microbiol 10, 538–550. https://doi.org/10.1038/nrmicro2832

Considering that the NGS does not produce quantitative data and that mosquitoes have not been analyzed with reference to specific tissues / organs, I believe it is deeply wrong to speak of interactions and networks.

Reply: One aim of this study was to verify possible bacterial interactions (co-exclusion /co-occurrence) in each mosquito species studied and then to observe potential differences between these interactions in the different mosquito species. It is known that variations in bacterial communities occur in different organs of the mosquitoes, but as there is no study in the literature with this type of analysis for mosquitoes collected in Vale do Ribeira, we believe that this study can guide future research, which can be targeted to specific organs of the invertebrate.

16S data have been used for observation of microbial co-exclusion/co-occurrence in different organisms (Faust et al., 2012; Faust and Raes, 2012; Hegde et al., 2018).

To make the above questions clearer to the reader, the following texts have been added:

“16S rRNA data have been used to verify microbial co-exclusion/co-occurrence in different organisms [19,28,29]. Relative abundance and/or presence/absence parameters are used to verify bacterial interactions and to infer bacterial networks.” (Lines 72-74).

“...(such as co-exclusion and co-occurrence)...” (Line 86).

 “ASV table was changed as described in Hegde et al. [19] and then it was used for the bacterial interaction (co-exclusion/co-occurrence) analyses.”(Lines 209-210).

“The non-discarded data were used to infer a bacterial interaction network, with the blue lines corresponding to positive correlation, whereas red lines to negative.” (Lines 230-232).

“...bacterial interaction analysis and then interaction data was used to infer bacterial interaction networks.” (Lines 324-325).

The discussion is really too verbose and in the first part it seems more like an introduction.

There are some spelling errors (i.e. lane 146 species instead of species)

Reply: Thank you for your valuable comment and suggestion. The text was revised and the spelling errors corrected.

“...each species...” (Line 214).

Reviewer 2 Report

The MS concerns microbial taxa related with 8 mosquito species collected in Southeastern São Paulo State, Brasil. MS doesn't have any serious flaws that would require major revisions. However, there are some shortcomings that should be corrected before it is published:

1. Please check the title: are you really mean about interaction network among microbial diversities? probably should read "Bacterial community diversity and bacterial interaction networks in eight mosquito species".

2. The sentence that “Mosquitoes harbour great amounts of fungi and bacteria.” (L29) is redundant.

3. In the sentence “The bacterial composition of the microbiota of these invertebrates depends on several factors, such as sex, larval habitat, developmental stage, environment, and species” (L30-31) add information that also can be related with microsporidian infection (Trzebny, A., Slodkowicz-Kowalska, A., Björkroth, J., & Dabert, M. (2021). Microsporidian Infection in Mosquitoes ( Culicidae ) Is Associated with Gut Microbiome Composition and Predicted Gut Microbiome Functional Content. Microbial Ecology, 0123456789. https://doi.org/10.1007/s00248-021-01944-z)

4. L52: there are more reports concerning microbiome changes in relation with not only viral, but also bacterial, microsporidian, fungal and Plasmodium; please check SCOPUS database with appropriate key words. (Some additional papers that can be included into the Introduction and/or the Discussion are listed at the end of this review).

5. L87, 89 – remove adapter sequences from the fusion primers and leave only the specific primer seq (e.g., GTGCCAGCMGCCGCGGTAA in Forward). Add references to the primers used in this study.

6. L136 - what R software packages were used? ggplot?

7. L146 – add “s” in species.

8. L147 – sequence reads (not “readings”), but this sentence should be rewritten (missing words).

9. L147 – this normalization approach seems strange; can you give a reference to justify its use?

10. L162 – this number does not agree with the number in the supplementary materials (Tab. S1).

11. L169 – compare the numbers with those in Tab. S2 (some differs).

12. Fig. 2 and the text: add information about the threshold used for the “remaining” taxa (gray bars are quite huge; did they contain numerous species?

13. L196-197 – there is no information about Dunn’s test adjusted with the Bonferroni method in Materials and methods section.

14. Fig. 3 and elsewhere in the text – generic Latin names of mosquitoes should be abbreviated to one letter (we use two letters only for different genera beginning with the same letter; this is not the case here).

15. L158-161 – Arrange species alphabetically or by abundance - they are now presented randomly.

16. Fig. 4 – The numbers in the abundance (heatmap legend) should be corrected (% instead of 1 – minus 3 range).

17. Supplementary Figure S6 is more informative than Fig. 5 in the body text. Please consider to give S6 instead of Fig. 5 and enlarge the new figure to make it more readable

18. It is a pity that the authors do not discuss in more depth the relationships they discovered between the dominant bacterial taxa and their possible influence on each other and on the host.

19. Table S7 has not been mentioned in the body text.

20. Below are some important papers that may be useful in this MS:

Caragata EP, Tikhe CV, Dimopoulos G (2019) Curious entanglements: interactions between mosquitoes, their microbiota, and arboviruses. Curr Opin Virol 37:26–36. https:// doi. org/ 10. 1016/j.coviro. 2019. 05. 005

Coon KL, Brown MR, Strand MR (2016) Mosquitoes host communities of bacteria that are essential for development but vary greatly between local habitats. Mol Ecol 25:5806–5826. https://doi. org/ 10. 1111/ mec. 13877

Gao H, Cui C, Wang L et al (2020) Mosquito microbiota and implications for disease control. Trends Parasitol 36:98–111. https:// doi. org/ 10. 1016/j. pt. 2019. 12. 001

Lee J-H, Lee K-A, Lee W-J (2017) Microbiota, gut physiology, and insect immunity. In: Ligoxygakis P (ed) Advances in Insect Physiology. Elsevier Ltd., pp 111–138. https:// doi. org/ 10. 1016/bs. aiip. 2016. 11. 001

Onchuru TO, Ajamma YU, Burugu M et al (2016) Chemical parameters and bacterial communities associated with larval habitats of Anopheles, Culex and Aedes mosquitoes (Diptera: Culicidae) in western Kenya. Int J Trop Insect Sci 36:146–160. https:// doi. org/ 10. 1017/ S1742 75841 60000 96

Scolari F, Casiraghi M, Bonizzoni M (2019) Aedes spp. and their microbiota: a review. Front Microbiol 10:e2036. https:// doi. org/10. 3389/ fmicb. 2019. 02036

Strand MR (2018) Composition and functional roles of the gut microbiota in mosquitoes. Curr Opin Insect Sci 28:59–65. https://doi. org/ 10. 1016/j. cois. 2018. 05. 008

Author Response

Revisor 2

The MS concerns microbial taxa related with 8 mosquito species collected in Southeastern São Paulo State, Brasil. MS doesn't have any serious flaws that would require major revisions. However, there are some shortcomings that should be corrected before it is published:

  1. Please check the title: are you really mean about interaction network among microbial diversities? probably should read "Bacterial community diversity and bacterial interaction networks in eight mosquito species".

Reply: The authors appreciate your time and effort in reviewing this manuscript. Thank you for this observation. We changed the title to “Bacterial community diversity and bacterial interaction networks in eight mosquito species” (Lines 2-3).

  1. The sentence that “Mosquitoes harbour great amounts of fungi and bacteria.” (L29) is redundant.

Reply: Thank you for this comment. The sentence “Mosquitoes harbour substantial amounts of fungi and bacteria” has been changed to “Mosquitoes harbour a great diversity of fungi and bacteria” (Lines 29-30).

  1. In the sentence “The bacterial composition of the microbiota of these invertebrates depends on several factors, such as sex, larval habitat, developmental stage, environment, and species” (L30-31) add information that also can be related with microsporidian infection (Trzebny, A., Slodkowicz-Kowalska, A., Björkroth, J., & Dabert, M. (2021). Microsporidian Infection in Mosquitoes ( Culicidae ) Is Associated with Gut Microbiome Composition and Predicted Gut Microbiome Functional Content. Microbial Ecology, 0123456789. https://doi.org/10.1007/s00248-021-01944-z)

Reply: We appreciate your suggestion and we added the requested information. Now it reads:

 “The bacterial composition of the microbiota of these invertebrates depends on several factors, such as sex, larval habitat, developmental stage, environment, species and can be related with microsporidian infection [4-7].” (Lines 30-32).

  1. L52: there are more reports concerning microbiome changes in relation with not only viral, but also bacterial, microsporidian, fungal and Plasmodium; please check SCOPUS database with appropriate key words. (Some additional papers that can be included into the Introduction and/or the Discussion are listed at the end of this review).

Reply: We appreciated your suggestions and the list of additional papers to be added to improve the manuscript. We added additional information and the text now reads:

“Interactions between bacteria, viruses, microsporidian, fungi and Plasmodium also changes the bacterial community [6,22-25]. Specimens of Culex pipiens infected with the West Nile virus showed an increased proportion of Serratia [26]. In addition, the endosymbiotic bacteria Spiroplasma sp. PL03 and Weissela cf. viridescens depends on microsporidia infection in the mosquito gut [6], and the bacterial microbiota was distinct between specimens of An. gambiae and Anopheles funestus that were positive and negative groups for Plasmodium falciparum [22]. In addition, in the same study, authors observed that some bacterial species such as Asaia borgorensis, Burkholderia fungorum, Burkholderia cepacia and Enterobacter cloacae were present only in females that were negative for P. falciparum. Balaji et al. [27] demonstrated that Wolbachia can influence the colonization of certain bacterial taxa by competitive interactions, such as the abundance of Serratia sp. in Wolbachia-carrying mosquitoes.” (Lines 60-71).

  1. L87, 89 – remove adapter sequences from the fusion primers and leave only the specific primer seq (e.g., GTGCCAGCMGCCGCGGTAA in Forward). Add references to the primers used in this study.

Reply: Thank you very much for the correction. The adapter sequences were removed and the primers’ reference added. It now reads:

“...(16S-V4 Forward: 5′ GTGCCAGCMGCCGCGGTAA 3′; 16S- V4 Reverse: 5′ GGACTACHVGGGTWTCTAAT 3′) [32]...” (Lines 146-148).

  1. L136 - what R software packages were used? ggplot?

Reply: We acknowledge you for inquiring about the program. We included the packages employed to develop the PCoA figures. The text “...perform Principal Coordinate Analysis (PCoA) in RStudio v.1.4.1106. PCoA images allow visualising...” was changed to “...perform Principal Coordinate Analysis (PCoA) in RStudio v.1.4.1106. PCoA images were generated with tidyverse and qiime2R packages available in RStudio. These images allow visualising...” (Lines 201-203).

  1. L146 – add “s” in species.

Reply: Thank you for showing our mistake. We added “s” in species and now it reads: “... for each species...” (Lines 213-214).

  1. L147 – sequence reads (not “readings”), but this sentence should be rewritten (missing words).

Reply: We greatly appreciated your suggestion to rewrite the text.

The text “The following modifications were performed to filter ASV table data: (1) ASVs with several readings lower than 0.1% of the total number of readings from all samples were removed; (2) The remaining ASVs were combined according to common taxonomy assignments lowest until genus level; (3) for each species, a relative abundance table was generated by dividing the number of readings of a given bacterial taxon against a total and an initial number of readings each sample and then multiplying by 100.” was rewritten and now it reads:

“We performed the modifications to filter ASV table data: (1) ASVs with readings lower than 0.1% of the total number of readings from all samples were removed; (2) The remaining ASVs were combined according to common taxonomy assignments lowest until genus level; (3) for each species, a relative abundance table was generated by dividing the number of sequence reads of each bacterial taxon per the total and initial number of sequence reads of each sample and then multiplying by 100.” (Lines 210-216).

  1. L147 – this normalization approach seems strange; can you give a reference to justify its use?

Reply: Thank you for this comment. The changed ASV table was used for bacterial interaction analyses. We performed changes in the table as described in Hegde et al. 2018 (https://doi.org/10.3389/fmicb.2018.02160). To make it clearer to the reader, we added the reference that we used to modify mentioned above. Now it reads:

“ASV table was changed as described in Hegde et al. [19] and then it was used for the bacterial interaction (co-exclusion/co-occurrence) analyses.”(Lines 209-210).

  1. L162 – this number does not agree with the number in the supplementary materials (Tab. S1).

Reply: Thank you for this comment. We verified the cited number both in the text and in table S1 and they are in agreement. A total of 9,647,379 raw reads were generated for both forward (R1) and reverse (R2) sequences and, in table S1, they correspond to the sum of column D to R1 (9,647,379) or E to R2 (9,647,379).

  1. L169 – compare the numbers with those in Tab. S2 (some differs).

Reply: We appreciate the observation. We compared the numbers and we did the modifications necessary. The text “...ASVs; Hg. leucocelaenus, 310; Hg. capricornii, 334; Ps. ferox, 874; Ae. serratus, 505; Sa. conditus,...” was changed to “...ASVs; Hg. leucocelaenus, 310; Hg. capricornii, 334; Ps. ferox, 874; Ae. serratus, 504; Sa. conditus,...” (Line 247).

  1. Fig. 2 and the text: add information about the threshold used for the “remaining” taxa (gray bars are quite huge; did they contain numerous species?

Reply: We appreciate the observation. In figure 2, “remainder” (gray bars) contain numerous genera and corresponds to genera that had 2% or less of the total contigs analyzed. We added this information in figure 2 legend and now it reads:  

“Figure 2. Bar graph depicting the bacterial amplicon sequence variants (ASV) composition at the bottom taxonomic level (genus) for each mosquito species. Remainder (gray bars) corresponds to genera that had 2% or less of the total contigs analyzed.” (Lines 271-273).

  1. L196-197 – there is no information about Dunn’s test adjusted with the Bonferroni method in Materials and methods section.

Reply: We appreciate the observation. We added the information about method used to adjust Dunn's test. So, the text “Shannon-Weaver indices (α diversity) were subject to Kruskal-Wallis followed Dunn test in RStudio v.1.4.1106...” was changed to “Shannon-Weaver indices (α diversity) were subject to Kruskal-Wallis followed by Dunn’s test adjusted with the Bonferroni method in RStudio v.1.4.1106...” (Lines 185-186).

  1. Fig. 3 and elsewhere in the text – generic Latin names of mosquitoes should be abbreviated to one letter (we use two letters only for different genera beginning with the same letter; this is not the case here).

Reply: Thank you for this comment. In 1975, Reinert proposed the 2-letter abbreviation for genera and 3-letter for subgenera of family Culicidae. Due to the occurrence of generic-level changes, revisions of the abbreviations were carried out by Reinert in other years, such as in 2001 and 2009. Numerous studies of Culicidae follow Reinert’s proposal and this study chose to adopt it as well and therefore the use of the two-letter abbreviation even when there is no different genera beginning with the same letter.

  1. L158-161 – Arrange species alphabetically or by abundance - they are now presented randomly.

Reply: We thank the reviewer for this suggestion. We arranged alphabetically the species. Now it reads:

“Aedes scapularis (08), Aedes serratus (06), Haemagogus capricornii (07), Haemagogus leucocelaenus (14), Kerteszia cruzii (15), Psorophora ferox (12), Sabethes conditus (30) and Wyeomyia confusa (19)...” (Lines 237-239).

  1. Fig. 4 – The numbers in the abundance (heatmap legend) should be corrected (% instead of 1 – minus 3 range).

Reply: Thank you for this comment. We redid figure 4 using relative abundance data for each sample. We changed the Figure 4 legend to:

“Figure 4. Heatmap of bacterial sequences with taxonomic assignment to genus level in each mosquito sample. Each row represents a bacterial taxon, and each column corresponds to a mosquito sample. Relative abundance data are assigned colours across a gradient, from yellow (higher bacterial abundance) to purple (lowest bacterial abundance). (A) corresponds to samples of the species Ae. scapularis; (B) corresponds to samples of the species Ae. serratus; (C) corresponds to samples of the species Hg. capricornii; (D) corresponds to samples of the species Hg. leucocelaenus; (E) corresponds to samples of the species Ke. cruzii; (F) corresponds to samples of the species Ps. ferox; (G) corresponds to samples of the species Sa. conditus and (H) corresponds to samples of the species Wy. confusa.” (Lines 311-319).

  1. Supplementary Figure S6 is more informative than Fig. 5 in the body text. Please consider to give S6 instead of Fig. 5 and enlarge the new figure to make it more readable

Reply: We thank the reviewer for this suggestion. We considered the Figure S6 instead of Figura 5. We removed the images corresponded Figure 5 and Figure S6 and added two new figures to show the bacterial interaction, Figures 5 and 6. Also modified the text of the Figure 5 legend and added the Figure 6 legend:

Figure 5. Bacterial interaction network in Ae. scapularis, Ae. serratus, Hg. capricornii and Hg. leucocelaenus. Each colored circle represents a bacterial taxon. Blue line corresponds to a positive correlation between taxa. Red line represents a negative correlation between taxa. Correlations were obtained by Spearman and only considered correlations with r > 0.75 and p > 0.05.” (Lines 391-394).

Figure 6. Bacterial interaction network in Ke. cruzii, Ps. ferox, Sa. conditus and Wy. confusa. Each colored circle represents a bacterial taxon. Blue line corresponds to a positive correlation between taxa. Red line represents a negative correlation between taxa. Correlations were obtained by Spearman and only considered correlations with r > 0.75 and p > 0.05.” (Lines 401-404).

  1. It is a pity that the authors do not discuss in more depth the relationships they discovered between the dominant bacterial taxa and their possible influence on each other and on the host.

Reply: We thank the comment. We added the following text about Wolbachia:

“ Wolbachia infection in Ae. aegypti causes upregulation in the transcription of genes related to reduction-oxidation reactions and immunity, leading to an increase of reactive oxygen species and induction of oxidative stress in the host and showing that to favor infection, the bacterium can manipulate the host’s defense system [64].

Many of Wolbachia strains induce cytoplasmic incompatibility (IC), resulting in lethality of embryos generated from females not infected with Wolbachia or infected with a different strain present in the male [65]. It manipulates the reproduction of the host and facilitates the spread of this bacterium, since infected females have this reproductive advantage over uninfected ones. Many studies use CI as a method of biological control [66]. In addition, this bacterium can reduce the infection of the dengue virus in the salivary glands of Ae. albopictus [67] and to inhibit yellow fever virus replication in Aedes aegypti [68]. Thus, considering that this bacterium was abundant in samples of Hg. leucocelaenus and this species of mosquito is considered the primary vector of sylvatic yellow fever virus in the south-eastern region of Brazil [38], further studies need to be carried out to know the Wolbachia strains present in these mosquitoes and to know if the existing strains can harm the spread of the yellow fever virus.” (Lines 503-531).

  1. Table S7 has not been mentioned in the body text.

Reply: Thank you for comment. We mentioned the table S7 in Results, line 330.

  1. Below are some important papers that may be useful in this MS:

Reply: We thank the reviewer for the suggestion. The papers below were cited in the manuscript.

Caragata EP, Tikhe CV, Dimopoulos G (2019) Curious entanglements: interactions between mosquitoes, their microbiota, and arboviruses. Curr Opin Virol 37:26–36. https:// doi. org/ 10. 1016/j.coviro. 2019. 05. 005

Coon KL, Brown MR, Strand MR (2016) Mosquitoes host communities of bacteria that are essential for development but vary greatly between local habitats. Mol Ecol 25:5806–5826. https://doi. org/ 10. 1111/ mec. 13877

Gao H, Cui C, Wang L et al (2020) Mosquito microbiota and implications for disease control. Trends Parasitol 36:98–111. https:// doi. org/ 10. 1016/j. pt. 2019. 12. 001

Scolari F, Casiraghi M, Bonizzoni M (2019) Aedes spp. and their microbiota: a review. Front Microbiol 10:e2036. https:// doi. org/10. 3389/ fmicb. 2019. 02036